# IFN-γ stimulated murine and human neurons mount anti-parasitic defenses against the intracellular parasite *Toxoplasma gondii*

Sambamurthy Chandrasekaran[1], Joshua A. Kochanowsky[2,6], Emily F. Merritt[2], Joseph S. Lagas[3], Ayesha Swannigan[4] & Anita A. Koshy ®[1,2,5] ✉

Dogma holds that *Toxoplasma gondii* persists in neurons because neurons cannot clear intracellular parasites, even with IFN-γ stimulation. As several recent studies questioned this idea, here we use primary murine neuronal cultures from wild type and transgenic mice in combination with IFN-γ stimulation and parental and transgenic parasites to reassess IFN-γ dependent neuronal clearance of intracellular parasites. We find that neurons respond to IFN-γ and that a subset of neurons clear intracellular parasites via immunity regulated GTPases. Whole neuron reconstructions from mice infected with parasites that trigger neuron GFP expression only after full invasion reveal that ~50% of these *T. gondii*-invaded neurons no longer harbor parasites. Finally, IFN-γ stimulated human pluripotent stem cell derived neurons show an ~50% decrease in parasite infection rate when compared to unstimulated cultures. This work highlights the capability of human and murine neurons to mount cytokine-dependent anti-*T. gondii* defense mechanisms in vitro and in vivo.

A select number of highly divergent intracellular microbes (e.g. measles virus, polio virus, *Toxoplasma gondii*) cause infections of the central nervous system (CNS). Though these microbes infect many cell types, in the CNS, neurons are often preferentially infected. One commonly cited reason for this neuron predominance is that neurons lack the ability to mount traditional cell-intrinsic immune responses[1–3]. For example, neurons have low baseline levels of MHC I and STAT1[1,4–6]. However, considerable evidence shows that neurons can respond to type I and type II interferons[4,5,7,8] and clear certain viral pathogens (e.g. Sindbis virus[9,10] and vesicular stomatitis virus[11]). Limited work has been done on the capabilities of neurons to clear non-viral pathogens.

*Toxoplasma gondii* is a protozoan parasite that naturally infects most warm-blooded animals, including humans and mice. In most immune competent hosts, *T. gondii* establishes a persistent or latent infection by switching from its fast growing lytic form (the tachyzoite) to its slow growing, encysting form (the bradyzoite). In humans and mice, the CNS is a major organ of persistence and neurons are the

principal cell in which *T. gondii* cysts are found[12–15]. IFN-γ is essential for control of *T. gondii* both systemically and in the CNS[16,17], in part through the activation of the immunity regulated GTPase system (IRGs) which is critical for parasite control in hematopoietic and non-hematopoietic cells[18–20]. Based upon these findings and in vitro studies showing that *T. gondii* readily invades murine astrocytes and neurons, but only IFN-γ-stimulated astrocytes—not IFN-γ-stimulated neurons—clear intracellular parasites[19,21–23], the model of CNS toxoplasmosis was that during natural infection parasites enter the CNS, invade both astrocytes and neurons, after which astrocytes kill the intracellular parasites, leaving the immunologically incompetent neuron as the host cell for the persistent, encysted form of the parasite.

Several recent findings have called this model into question. Pan-cellular ectopic expression of an MHC I allele (H-2 L^d) associated with low levels of CNS persistence[24,25] leads to a lower CNS parasite burden than when mice lack expression of this MHC I allele in neurons only[26]. Moreover, the use of a Cre-based system that permanently marks CNS

[1]BIO5 Institute, University of Arizona, Tucson, AZ, USA. [2]Department of Immunobiology, University of Arizona, Tucson, AZ, USA. [3]Undergraduate Biology Research Program, University of Arizona, Tucson, AZ, USA. [4]Undergraduate Research Opportunities Consortium, University of Arizona, Tucson, AZ, USA. [5]Department of Neurology, University of Arizona, Tucson, AZ, USA. [6]Present address: University of California, Los Angeles, CA, USA. ✉e-mail: akoshy@arizona.edu

cells that have been injected with *T. gondii* proteins[27,28], revealed that parasites extensively interact with neurons and that the majority (>90%) of these *T. gondii*-injected neurons do not actively harbor cysts[15,27]. Together these in vivo studies question the prior model by raising the possibility that neurons clear intracellular parasites.

In this work, we use primary murine neuronal cultures from wild type and genetically modified mice in combination with cytokine stimulation and parental and transgenic parasites, including a new engineered *T. gondii*-Cre line, to reassess the ability of neurons to clear intracellular parasites in the setting of IFN-γ stimulation. Given the prior conflicting in vitro and in vivo findings, our data reveal that neurons respond to IFN-γ, including up-regulating the IRGs, and that a subset of neurons (~20%) clear intracellular parasites via the IRGs. In addition, in Cre reporter mice infected with *T. gondii*-Cre parasites that mark CNS cells only after full invasion, whole neuron reconstructions show that ~50% of these *T. gondii*-invaded neurons no longer harbor parasites. Finally, IFN-γ stimulation of human pluripotent stem cell derived neurons (hPSC-neurons) leads to an ~50% decrease in parasite infection rate when compared to unstimulated, infected cultures. Collectively, these data highly suggest that IFN-γ stimulation leads to parasite resistance in murine and human neurons and that a subset of murine neurons clear intracellular parasites both in vitro and in vivo, likely via the IRGs.

## Results
### IFN-γ stimulated primary pure murine cortical neurons show classical IFN-γ responses
As *T. gondii* primarily infects and encysts in the cortex[29–31], we sought to determine the response of cortical neurons to IFN-γ stimulation. To accomplish this goal, we exposed pure primary murine cortical neuronal cultures to 100 U/ml of IFN-γ or vehicle control for 4 and 24 h, followed by harvesting of total RNA. We chose these time points because prior work showed that hippocampal neurons have a delayed IFN-γ response[5]. After harvesting the RNA, we used quantitative real time-PCR (qRT-PCR) to quantify transcript levels of traditional IFN-γ-response genes (*STAT1*, *IRF1*, *MHC-I*) as well as the effector components of the IRG system (*Irga6*, *Irgb6*, and *Gbp2*)[32,33]. We found that *STAT1*, the classical transcription factor that drives the expression of many IFN-γ response genes, and *IRF1* were highly upregulated (4 h: 3–5 $\log_2$ fold; 24 h: 5–7 $\log_2$ fold) compared to unstimulated neurons (Fig. 1A), while MHC-I showed a more modest level of upregulation (~ 2 $\log_2$ fold) and only at 24 h. In addition, consistent with findings in non-neuronal murine cells, compared to unstimulated neurons, IFN-γ stimulated neurons also significantly up-regulated *Irga6*, *Irgb6*, and *Gbp2* (4 h: 4–7 $\log_2$ folds; 24 h: 7–9 $\log_2$ folds) (Fig. 1A)[34–36]. To determine how these increased transcript levels translated to protein levels, we isolated total protein lysates from unstimulated and IFN-γ stimulated cultures. For STAT1, we blotted both for total STAT1 and for phosphorylated STAT1 (pSTAT1), the active form that enters the nucleus and binds DNA. Compared to unstimulated cultures, IFN-γ stimulated cultures showed a > 10-fold increase in protein levels for total STAT1 and pSTAT1 at 24 h post-stimulation and a > 35-fold increase at 48 h post-stimulation (Fig. 1B, C). The undetectable level of total STAT1 in unstimulated neurons is consistent with previously published data suggesting that neurons have low or no constitutive amounts of STAT1[37]. Similarly, at 24 h post-stimulation, the IRG complex effector proteins Irga6 and Irgb6 increased an ~7-fold and 10-fold respectively over unstimulated cultures, a level that was maintained at 48 h post stimulation (Fig. 1B, C).

To confirm that the detected changes were primarily driven by neurons and not by glial cells that commonly cause low levels of contamination in "pure" neuronal cultures, we stained the cultures to determine what percentage of cells were neurons, astrocytes, or microglia. These analyses showed that our cultures were consistently 95% neurons and 5% astrocytes; no microglia were observed

(Supplementary Fig. 1). To determine the level of nuclear translocation of pSTAT1 at the single neuron level, we stained unstimulated and stimulated (100 U/ml for 24 h) primary neuron cultures with DAPI and antibodies against Tuj1, a neuronal marker, and pSTAT1. We then quantified the pSTAT1 signal intensity in Tuj1[+] nuclei. In unstimulated cultures, we observed very little pSTAT1 signal in Tuj1[+] nuclei (MFI 2236 ± 353.24, mean ± SEM) (Fig. 1D). Conversely, IFN-γ stimulated neurons showed an ~10–90-fold increase in pSTAT1 staining in Tuj1[+] nuclei (Fig. 1D).

Together these data show that IFN-γ stimulated primary murine cortical neuron cultures upregulate IFN-γ response genes and proteins in a delayed manner consistent with what has been observed in hippocampal neurons[5]. The upregulated genes and proteins include *STAT1* and the IRG genes known to be required for IFN-γ-dependent killing of intracellular parasites in murine non-neuronal cells[33]. Collectively, these data suggest that IFN-γ stimulated murine cortical neurons upregulate the appropriate machinery to clear intracellular parasites via the IRG system.

### IFN-γ pre-stimulation leads to a decrease in the percentage of *T. gondii*-infected neurons
To address neuronal capability for clearing intracellular parasites, we infected IFN-γ stimulated or unstimulated neurons with representative strains from the three genetically distinct canonical *T. gondii* strain types (type I (RH), II (Prugniaud), and III (CEP)), which differ in acute virulence in mice[38,39]. We quantified neuron infection rates at 3, 12, and 24 h post infection (hpi). At 3 hpi, stimulated and unstimulated cultures showed a similar infection rate, regardless of infecting strain (Fig. 2A–C). At 12 and 24 hpi, the type I strain continued to show the same infection rate in both vehicle and IFN-γ stimulated cultures (Fig. 2A). Conversely, IFN-γ stimulated cultures infected with type II or type III parasites showed an ~25% decrease in infection rate at 12 and 24 hpi, (Fig. 2A–C).

### GCre-expressing parasites show that neurons clear parasites in response to IFN- γ
While the prior data suggested that IFN-γ stimulated neurons might clear intracellular parasites in a strain-specific manner, these findings could also be explained by decreased rates of late invasion by type II and type III parasites in IFN-γ stimulated neurons. A late invasion defect is possible because clearance assays in neurons are limited by the inability to synchronize infection or vigorously wash off uninvaded parasites as either procedure causes widespread neuronal death. To distinguish between a late invasion defect and true clearance of intracellular parasites, we needed a way to specifically track neurons that were infected and subsequently cleared the intracellular parasite. As noted above, we had previously developed a Cre-based system that leads to GFP expression only in host cells injected with *T. gondii* proteins. In this system, Cre is fused to toxofilin, a rhoptry protein (ROP). ROPs are parasite proteins that are injected into host cells *prior to invasion*, which means that parasite-triggered host cell expression of GFP does not require parasite invasion (i.e., aborted invasion) (Fig. 2D). Thus, RCre (ROP::Cre)-expressing parasites do not help us distinguish between increased aborted invasion versus clearance of intracellular parasites. One way to address this issue would be to fuse Cre to GRA16, a dense granule protein (GRA) that is released into host cells only *after* invasion[40–43]. Thus, GCre (GRA::Cre)-expressing parasites should not cause Cre-mediated recombination in the setting of aborted invasion and should only identify host cells that were or had previously been infected (Fig. 2D).

To test this potential solution, we engineered type II parasites to express an HA-tagged GCre (II-GCre). Using immunofluorescent assays and plaque assays, we determined that GCre was expressed, did not localize to the rhoptries, and that the expression of GCre did not affect overall parasite viability (Supplementary Fig. 2). To test the capability

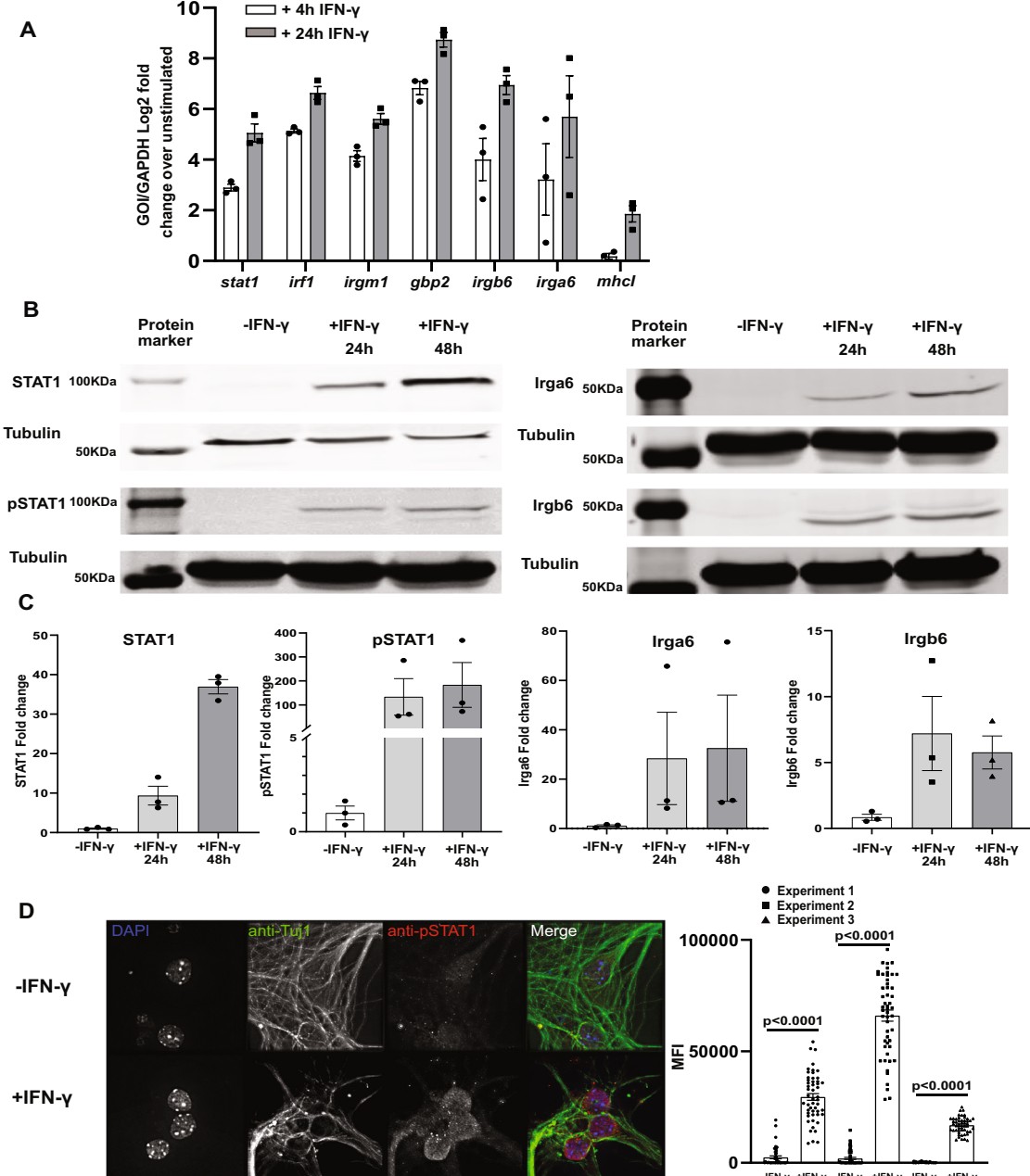

**Fig. 1 | IFN-γ stimulated primary murine neurons show intact IFN-γ signaling pathway and express genes involved in *T. gondii* clearance.** Primary neurons cultures were stimulated with vehicle or IFN-γ (100 U/ml), followed by RNA or protein extraction or immunofluorescent assays. **A** Quantification of specified genes using qRT-PCR. **B** Representative images of western blots for specified proteins from unstimulated and IFN-γ stimulated cultures. **C** Densitometric quantification of western blots. Proteins of interest are normalized to β-tubulin and then shown as a fold change over unstimulated cultures. For **A**, **C** Bars, mean ± SEM. *N* = 3

independent experiments. **D** *Images*: Representative images of unstimulated or IFN-γ stimulated neurons stained as indicated. *Merge image*: blue = DAPI, green = anti-Tuj1 (anti-neuron antibody) = green, red = anti-pSTAT. Scale bar = 5 μm. *Graph*: Quantification of the mean fluorescent intensity (MFI) of pSTAT1 nuclear signal in unstimulated or IFN-γ stimulated Tuj1⁺ cells. *N* = 48–51 nuclei/experiment, 3 independent experiments. Bars, mean ± SD. Unpaired t-test with Welch's correction. **A–C, D graph** Source data are provided as a Source Data file.

of II-GCre parasites to trigger Cre-mediated recombination, we infected Cre reporter fibroblasts that express GFP only after Cre-mediated recombination[28] with II-RCre parasites, II-GCre parasites, or parental type II parasites (no Cre expression). At 24 hpi, we observed that both II-RCre parasites and II-GCre parasites caused host cell expression of GFP, while the parental strain did not (Supplementary Fig. 3A, B). Compared to II-RCre parasites, II-GCre parasites were less efficient at triggering Cre-mediated recombination (>90 vs. 50%, Supplementary

Fig. 3B), which was expected because host cell-exported GRA proteins show decreased exportation when fused to ordered proteins[41,44,45].

Having confirmed that II-GCre parasites trigger Cre-mediated recombination, we next assessed the capability of II-GCre parasites to identify only infected cells. To test this possibility, we infected cortical neuron cultures from Cre reporter mice for 72 h, which was the time point when we saw robust neuronal GFP expression. In II-RCre infected cultures, 67 ± 1.82% of GFP⁺ neurons were infected. In II-GCre infected

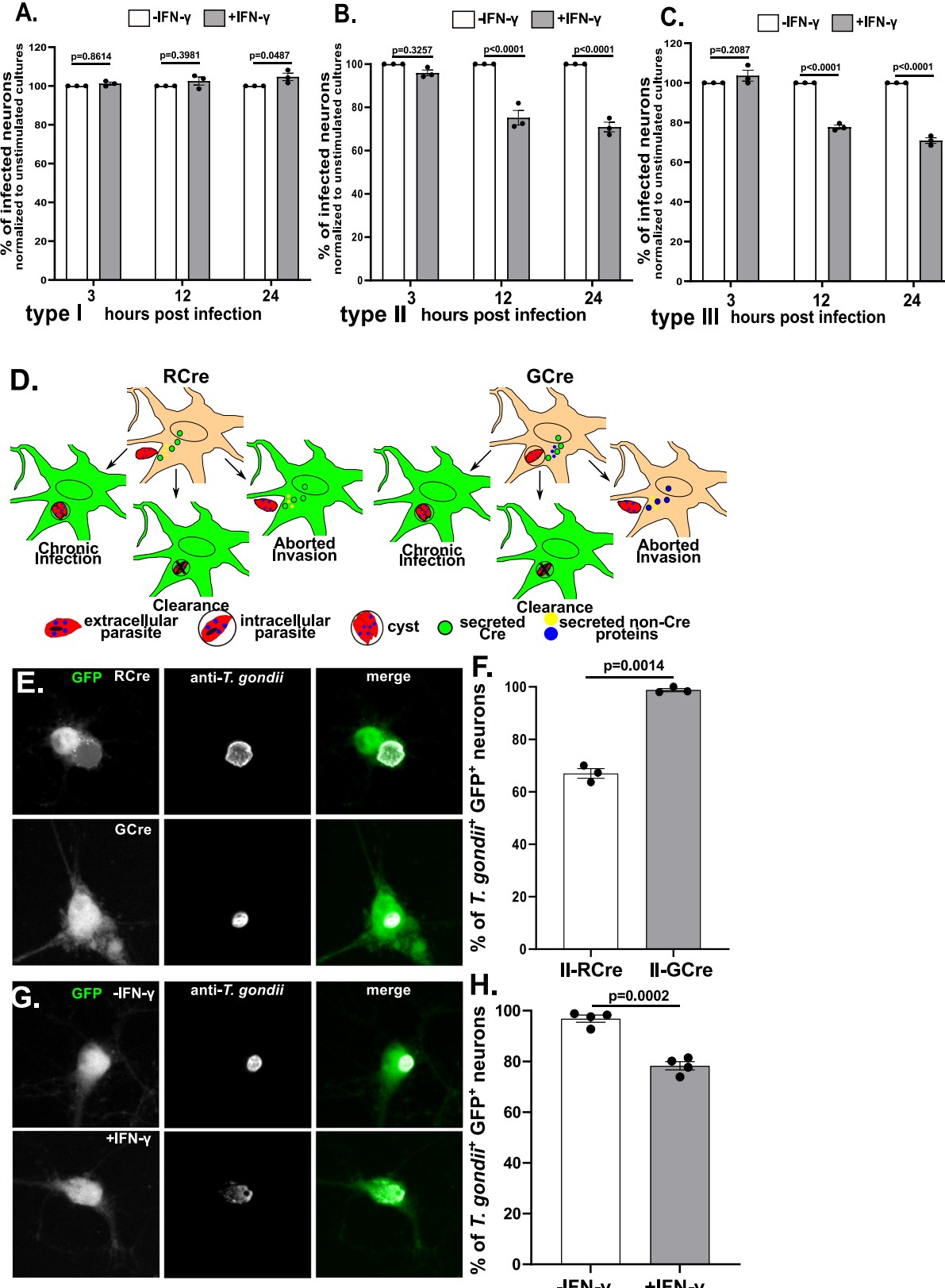

cultures, 98 ± 0.62% of GFP⁺ neurons were infected (Fig. 2E, F). As cultured neurons do not divide, we assume the ~30% of uninfected GFP⁺ neurons in II-RCre infected cultures arise from aborted invasion (Fig. 2F). More importantly, that ~100% of GFP⁺ neurons are infected when using II-GCre parasites confirms that II-GCre parasites trigger Cre-mediated recombination only after fully invading the host cell.

Having engineered parasites that only identify fully infected neurons, we next tested how IFN-γ pre-stimulation affected the rate of infection of GFP⁺ neurons by stimulating Cre reporter neuron cultures with vehicle or IFN-γ (100 U/ml) for 24 h prior to infection with II-GCre parasites. Consistent with previous results (Fig. 2F), in the vehicle treated cultures, 97 ± 1.4% of GFP⁺ neurons harbored a parasite

**Fig. 2 | IFN-γ stimulated neurons clear intracellular parasites.** Primary neuron cultures were treated as in Fig. 1. **A** Graph of the percentage of infected neurons, normalized to the unstimulated culture, for type I parasites. **B** As in **A** except for type II parasites. **C** As in **A** except for type III parasites. **A**–**C** Bars mean ± SEM. *N* = 759–1538 neurons/*T. gondii* strain/experiment, 3 independent experiments. 2-way ANOVA with Sidak's multiple comparisons. **D** Schematic of host cells labeled by RCre versus GCre parasites. *Left*, With RCre parasites the Cre fusion protein is secreted prior to invasion, leading to GFP-expressing neurons that arise from: (i) injection, invasion, and chronic infection, (ii) injection and invasion followed by neuronal clearance of the parasite, and (iii) injection of the protein *without* invasion (aborted invasion). *Right*, with GCre parasites the Cre fusion protein is secreted post invasion, leading to GFP-expressing neurons that arise from: (i) invasion and

persistent infection or (ii) invasion followed by clearance of the parasite, but not from aborted invasion. **E** Representative images of GFP⁺ neurons infected with listed parasites. *Merge image*: green = GFP, white = parasites stained with anti-*T. gondii* antibodies. Scale bar = 10 μm. **F** Graph of the percentage of infected GFP⁺ neurons at 72 hpi (unstimulated cultures). *N* = 300 neurons/experiment, 3 independent experiments. **G** Representative images of unstimulated or IFN-γ stimulated primary neurons infected with II-GCre parasites at 72 h. *Merge image*, as in **E**. Scale bar = 10 μm. **H** Graph of the percentage of infected GFP⁺ neurons at 72 hpi. *N* = 290–452 neurons/experiment, 4 independent experiments. **F**, **H** Bars mean ± SEM. Unpaired t-test with Welch's correction. **A**–**C**, **F**, **H** Source data are provided as a Source Data file.

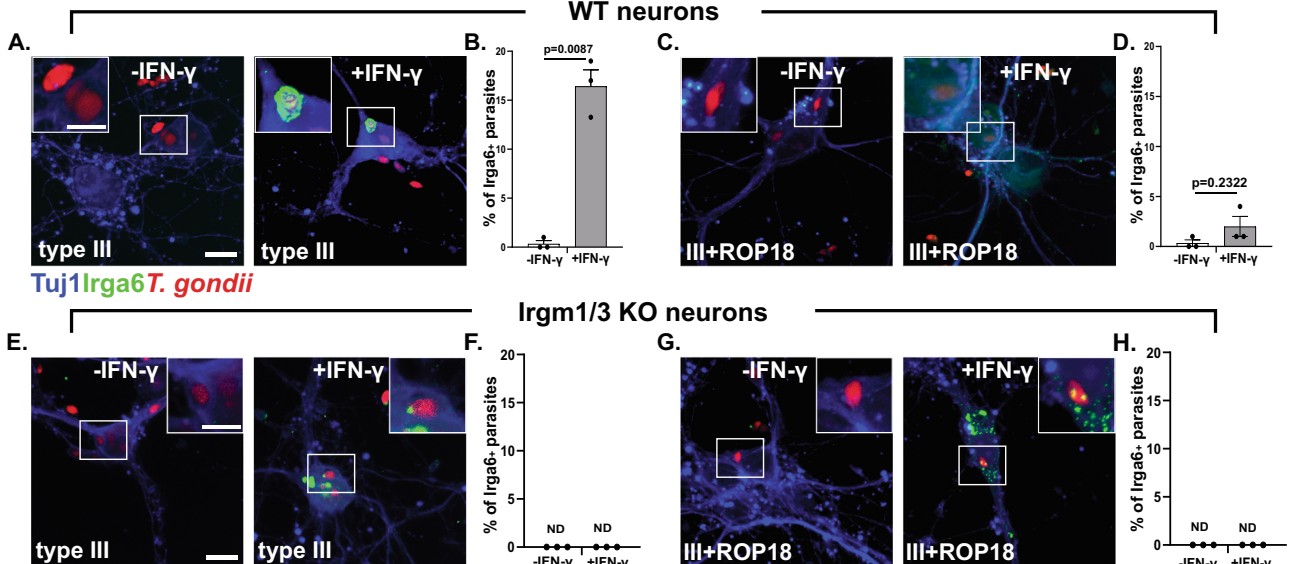

**Fig. 3 | IFN-γ stimulated wild type murine show increased loading of Irga6 onto the PVM of type III parasites.** Primary neuron cultures were treated as in Fig. 1, followed by infection with listed *T. gondii* strains. At 12 hpi cultures were fixed and stained with anti-Irga6 and anti-Tuj1 antibodies. The stained cultures were analyzed by confocal microscope. **A** Representative images of stained cultures from wild type (WT) mice infected with type III parasites. Blue = anti-Tuj1 antibodies, green = anti-Irga6 antibodies, red = mCherry expressing parasites. **B** Quantification of the percentage of Irga6⁺ parasitophorous vacuoles (PVs) in the setting of vehicle

or IFN-γ pre-stimulation. **C** Representative images as in **A** except infected with III + ROP18 parasites. **D** As in **B**. **E** Representative images as in **A** except using cultures from Irgm1/3 KO mice. **A**, **E** Scale bar = 10 μm full image, 5 μm inset. **F** As in **B**. **G** Representative images as in **C** except using Irgm1/3 KO neurons. **H** As in **B**. **B**, **D**, **F**, **H** Bars- mean ± SEM, *N* = 100–200 PVs/experiment, 3 independent experiments. ND = not detected. Unpaired t-test with Welch's correction. Source data are provided as a Source Data file.

(Fig. 2G, H). Conversely, in the setting of pre-treatment with IFN-γ, now only 78 ± 1.6% GFP⁺ neurons harbored parasites. These data suggest that the decrease in the rate of infection with type II and type III parasites in IFN-γ stimulated neurons (2B, C) is primarily mediated by IFN-γ stimulated neurons clearing intracellular parasites, rather than IFN-γ stimulation leading to decreased parasite invasion.

**In IFN-γ stimulated murine neurons Irga6 loads onto the PVM in a Irgm1/3-dependent manner**

As IFN-γ stimulated neurons up-regulate the IRG effectors (Fig. 1) and a portion clear IRG-sensitive intracellular parasites (Fig. 2), we next sought to determine if neurons use the IRG system to mediate IFN-γ-dependent killing of *T. gondii*. As loading of IRGs onto *T. gondii*'s parasitophorous vacuolar membrane (PVM) is indispensable for parasite clearance in IFN-γ stimulated non-neuronal cells[46] and as Irga6 is one of the effectors that loads onto the PVM[33], we analyzed the percentage of parasitophorous vacuoles (PVs) that were also Irga6⁺ in unstimulated and IFN-γ stimulated neurons infected with type III parasites. Consistent with findings in non-neuronal murine cells, we found that in unstimulated neurons, type III parasite PVs showed almost no Irga6 loading, while PVs in IFN-γ stimulated neurons showed an 8-fold increase in Irga6 loading (Fig. 3A, B). To further confirm these

findings, we also used a type III "IRG-resistant" strain[15]. This strain is engineered to express high levels of ROP18 (III + ROP18). ROP18 is a *T. gondii* kinase that phosphorylates Irga6 thereby preventing PV loading and effector oligomerization[47]. The parental type III strain has minimal expression of ROP18 because of an insertion in the promoter region of the *rop18* gene; it is this lack of ROP18 expression that renders the parental strain susceptible to the IRGs[48,49]. In cultures infected with III + ROP18 parasites, as expected, we now found almost no Irga6⁺ PVs even in the setting of IFN-γ stimulation (Fig. 3C, D). Finally, to confirm that IRG-loading followed the same principles in neurons as in non-neuronal cells, we used neurons from mice that lack the regulatory IRG components (*irgm1* and *irgm3*)[50], which for simplicity we will call Irgm1/3 KO mice. These regulatory components tether the effector components (e.g. Irga6) to the appropriate organelle until triggered to release the effectors onto the PV[51]; in cells that lack these proteins, the effectors inappropriately load onto host cell organelles and no longer target the PV[52]. As expected, in Irgm1/3 KO neurons, we found an increase in Irga6 aggregates in the setting of IFN-γ stimulation but no specific loading onto PVs regardless of IFN-γ stimulation or infecting strain (Fig. 3E–H).

Together, these data demonstrate that neuronal Irga6 loads onto the PVM of intracellular IRG-sensitive parasites in the setting of IFN-γ

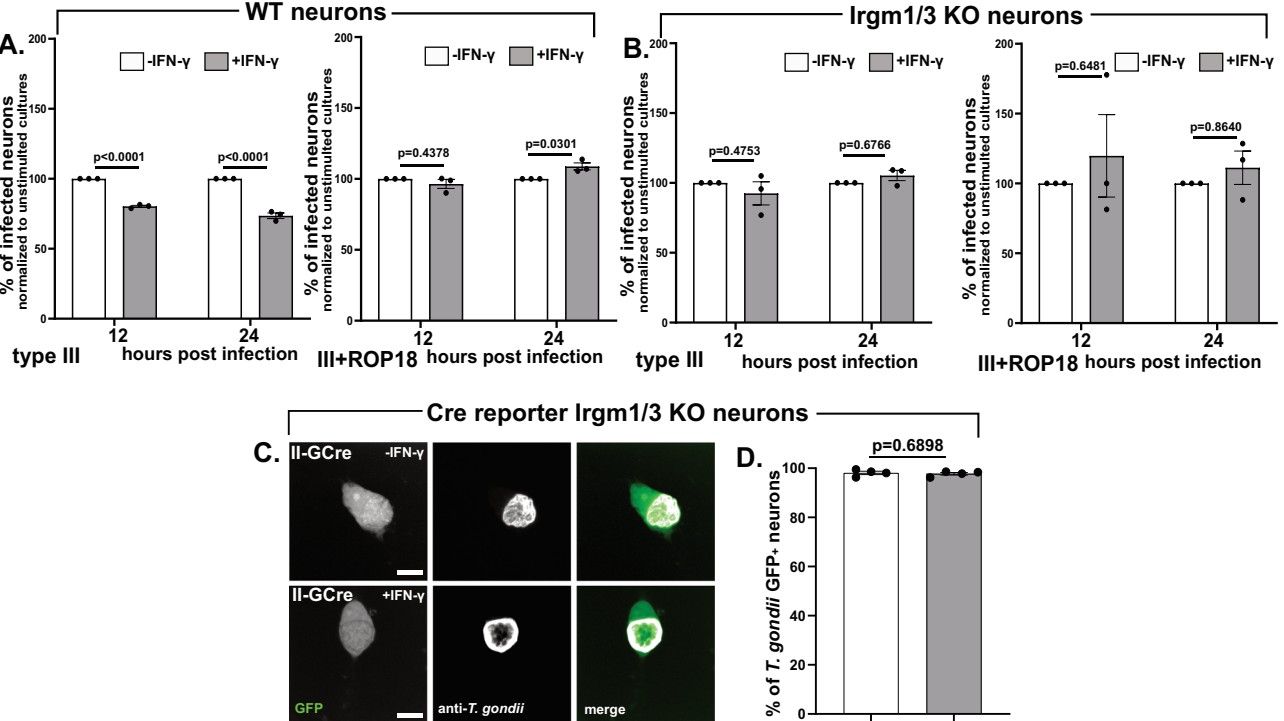

**Fig. 4 | An intact IRG system is required for IFN-γ-dependent murine neurons clearance of intracellular parasites.** Primary neuron cultures from WT (Cre reporter), Irgm1/3 KO, or Cre reporter Irgm1/3 KO mice were treated as in Fig. 1 followed by infection with listed *T. gondii* strains. **A** Graph of the percentage of infected WT neurons at listed time points for type III parasites (*left graph*) and III + ROP18 parasites (*right graph*), normalized to the unstimulated culture. **B** As in **A** except using Irgm1/3 KO neurons and either type III parasites (*left graph*) or III + ROP18 parasites (*right graph*). **A**, **B** At the listed time points, cultures were fixed, stained, and analyzed as in Fig. 2A. Bars mean ± SEM., *N* = 731–1448 neurons/*T.* *gondii* strain/experiment, 3 independent experiments. 2-way ANOVA with Sidak's multiple comparisons. **C** Representative images of GFP⁺ neurons from Cre reporter Irgm1/3 KO mice infected with II-GCre parasites. *Merge image*: Green = GFP, white = parasites stained with anti-*T. gondii* antibodies. Scale bars = 10 μm. **D** Graph of the percentage of infected GFP⁺ neurons. Bars, mean ± SEM. *N* = 389–506 neurons analyzed/experiment, 4 independent experiments. Unpaired t-test with Welch's correction. **C**, **D** Cultures were infected, fixed, stained, and analyzed as in Fig. 2F, G. **A**, **B**, **D** Source data are provided as a Source Data file.

pre-stimulation and when neurons have an intact IRG-system. These data strongly suggest that murine neurons use the IRG system for IFN-γ-dependent clearance/killing of intracellular parasites.

### IFN-γ stimulated murine neurons kill intracellular parasites using the IRG system

While the preceding data strongly suggests that murine neurons deploy the IRG-system to kill intracellular parasites in the settling of IFN-γ stimulation, they do not show that the IRG-system is essential for neuronal killing. To directly test this possibility, we took several approaches. First, we tested the ability of IFN-γ stimulated, wild type neurons to clear III + ROP18 parasites which show no Irga6⁺ loading even in the setting of IFN-γ stimulation (Fig. 3C, D). Indeed, in murine neuronal cultures infected with III + ROP18 parasites, we found the same rate of infection over time, regardless of IFN-γ stimulation (Fig. 4A). Second, we infected Irgm1/3 KO neuronal cultures with parental (type III, IRG-sensitive) or III + ROP18 parasites (IRG-resistant) and with or without IFN-γ pre-stimulation. We again found the same rate of infection, regardless of what strain we utilized (IRG-sensitive or resistant) and IFN-γ pre-stimulation state (Fig. 4B). As a final method of confirming that the lack of IRGs specifically affected intracellular parasites, we bred Irgm1/3 KO mice to Cre reporter mice to yield mice homozygous for the Cre reporter construct and that lack both *Irgm1* and *Irgm3*. We then generated neuronal cultures, which we stimulated with vehicle or IFN-γ, followed by infection with II-GCre parasites. In these cultures, ~98% (-IFN-γ: 98.12 ± 0.67%; +IFN-γ: 97.8% ± 0.53) of GFP⁺ neurons were infected, regardless of IFN-γ stimulation status (Fig. 4C, D).

Collectively, these data definitively show that in vitro IFN-γ stimulated neurons to kill intracellular IRG-sensitive parasites via the IRG system.

### Neurons clear intracellular parasites in vivo

Having shown that IFN-γ stimulated neurons clear intracellular parasites in vitro, we sought to determine if neurons cleared parasites in vivo. As II-GCre parasites trigger Cre-mediated recombination only after full host cell invasion (Fig. 2C), we reasoned that if we found GFP⁺, parasite⁻ neurons in Cre reporter mice infected with II-GCre parasites, these neurons must have cleared the invading parasite. To assess for GFP⁺ parasite⁻ neurons in vivo, we created whole neuron reconstructions from Hoechst-stained brain sections from 21 day post-infection (dpi) II-GCre infected mice[53,54] (Fig. 5A, B). Within these GFP⁺ neurons, we looked for areas that lacked GFP (Fig. 5B) because the PV excludes host cell GFP. We then confirmed that these areas contained parasites using Hoechst staining of parasite DNA (Fig. 5C, Supplementary Fig. 4). To confirm that our method for identifying parasites was accurate, for a subset of brain sections, we also stained the sections with either of two different anti-*T. gondii* antibodies. As expected, when we identified areas lacking GFP and showing Hoechst-stained parasite DNA, these areas were also positive for anti-*T. gondii* antibody staining (Fig. 5D, Supplementary Fig. 4A, B). Out of 35 reconstructed GFP⁺ neurons, we found that 17 (~50%) showed no evidence of persistent parasite infection (Fig. 5E). These data highly suggest that, in vivo, a substantial percentage of infected murine neurons clear intracellular parasites.

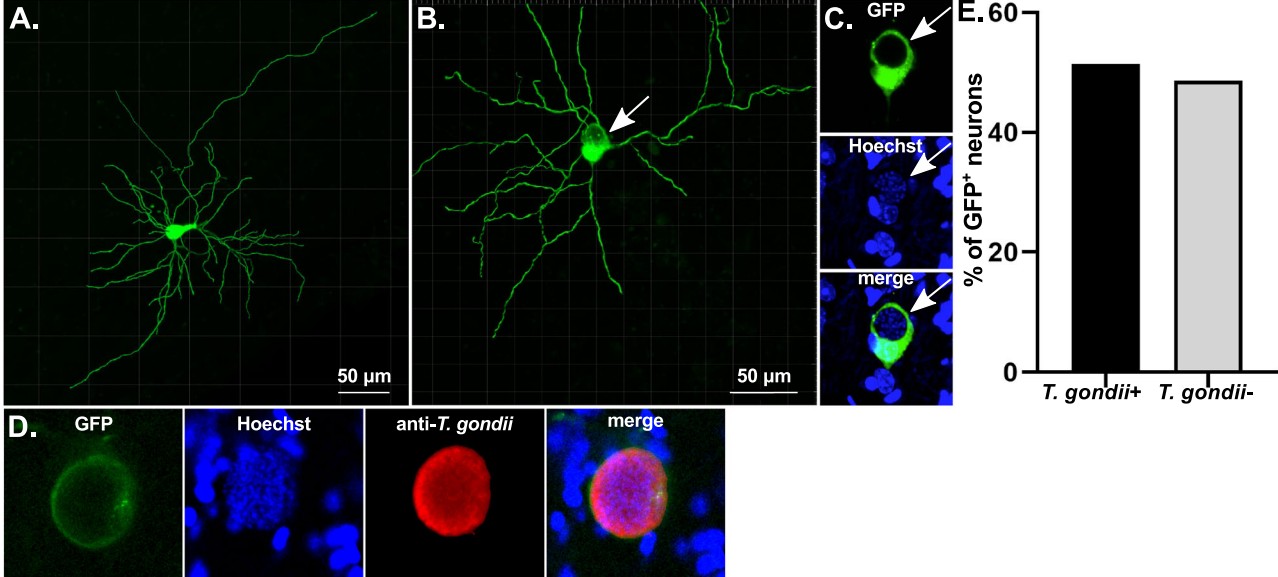

**Fig. 5 | Fifty percent of GFP⁺ neurons in mice infected with GCre parasites do not harbor parasites.** Cre reporter mice were infected with II-GCre parasites. At 21 dpi, brains were harvested and sagittally sectioned into 200 μm thick sections. Sections were cleared, stained with Hoechst ± anti-*T. gondii* antibodies, and imaged at 40× on a confocal microscope. Neurons in resulting images were rendered using Imaris software. **A** Representative rendering of a GFP⁺ neuron in which no parasites were identified. **B** As in **A** except now with a GFP⁺ neuron in which parasites were identified. White arrow shows parasites within neuron soma. **C** Single plane of soma from **B** *top image*: green = GFP, *middle image*: blue = Hoechst, *bottom image*: merge. Note the GFP displacement, suggesting parasite presence within the neuron, which is then confirmed by visualization of parasite nuclei stained with Hoechst (blue). **D** Representative image of a single plane of a rendering from a section stained with anti-*T. gondii* antibodies. Note that the parasite nuclei identified by Hoechst staining co-localize with the anti-*T. gondii* antibody staining. Green = GFP, blue = Hoechst, red = anti-*T. gondii* antibodies. **E** Graph of the percentage of rendered GFP⁺ neurons containing parasites (*T. gondii⁺*/infected) or not containing parasites (*T. gondii⁻*/uninfected). *N* = 35 neurons from 5 mice. Source data are provided as a Source Data file.

## IFN-γ stimulated human neurons show resistance to *T. gondii* infection

Having shown that murine neurons clear a portion of intracellular parasites in vitro and in vivo, we sought to translate these findings to human neurons. While mice and murine cells are good models for human infection (both are naturally infected with *T. gondii*, have the CNS as a major organ of persistence, have neurons as the major host cell for cysts, and require IFN-γ and CD8 T cells to control toxoplasmosis), differences exist between the two. In the current context, the most relevant difference is that human cells lack the expansive range of IRGs that mice have and instead rely on alternative mechanisms, which can vary by cell type, for IFN-γ-dependent control of *T. gondii*[55]. A prior study has shown that when IFN-γ stimulated human stem cell (hSC)-derived neurons are co-cultured with infected monocytes, parasite luciferase expression decreases compared to co-cultures of infected monocytes and unstimulated hSC-derived neurons[56]. As this work did not determine what caused the decrease in parasite luciferase expression, it is impossible to determine if the IFN-γ responses were being driven by effects on the neurons, infected monocytes, or both. Thus, we sought to more directly determine how IFN-γ stimulation influenced control of *T. gondii* in human neurons. To accomplish this goal, we derived neurons from human neuroprogenitor cells reprogrammed from an embryonic pluripotent stem cell line. After confirming that the hPSC neurons expressed appropriate cortical neuronal markers (Supplementary Fig. 5), we pre-treated hPSC neurons with human IFN-γ for 24 h, followed by infection with type II or III parasites. We then quantified the neuron infection rate at 3, 12, and 24 hpi via epifluorescent microscopy (Fig. 6). At 3 hpi, regardless of infecting strain, we found equivalent neuron infection rates between unstimulated and IFN-γ stimulated cultures. At 12 and 24 hpi, IFN-γ stimulated cultures showed a ≥ 50% decrease in the neuron infection rate compared to unstimulated cultures (Fig. 6). These data highly suggest that human neurons also mount anti-parasitic defenses in response to IFN-γ stimulation.

## Discussion

In this study, we sought to address the question: can neurons clear intracellular parasites? Using in vitro primary murine neurons from wild type and transgenic mice in combination with IFN-γ pre-stimulation and transgenic parasites, this work shows that a portion of murine neurons can and do clear intracellular parasites in an IFN-γ-dependent, IRG-dependent manner. Using our new II-GCre parasites, which trigger host cell mediated recombination only after full invasion, we also show that ~50% of neurons clear parasites in vivo. Finally, using hPSC neurons, we translated our findings to human neurons, showing that IFN-γ pre-stimulation decreases the infection rate by ~50%.

The data presented are the first to show that IFN-γ pre-stimulation enables human and murine neurons to partially resist successful infection by an intracellular eukaryotic pathogen. Importantly, in murine neurons, we leveraged *T. gondii* biology to show that this resistance was secondary to clearance of intracellular parasites (Fig. 2H), not simply an invasion defect (an important distinction in cultures where procedures such as aggressively washing off extracellular parasites cannot be done). In addition, our finding that IFN-γ stimulated murine neurons clear parasites in an IRG-dependent manner explains why a prior study found that IFN-γ stimulated primary murine neurons failed to clear intracellular parasites. The prior study, which was done at a time when neither the IRGs nor the parasite mechanisms to block the IRGs had been fully described, used a type I strain (RH) that we now know is IRG-resistant[23,47]. Consistent with the possibility that the use of this strain led to the negative findings, we also found no evidence for IFN-γ-dependent neuronal clearance when we infected with this IRG-resistant strain (Fig. 2A). In addition to addressing this prior work, our findings also potentially offer a mechanism for the recent work showing that MHC class I expression

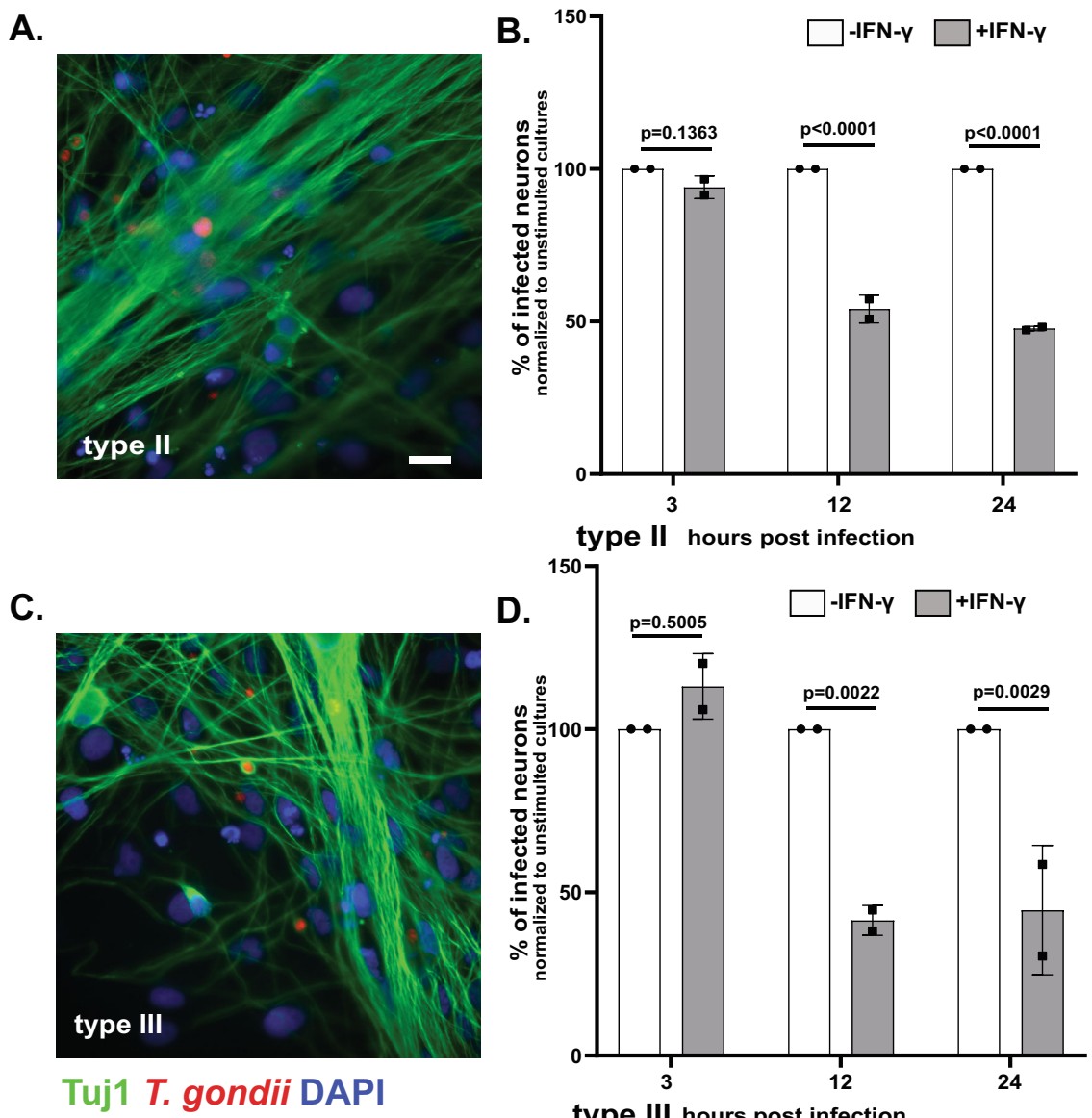

Tuj1 *T. gondii* DAPI

**Fig. 6 | IFN-γ stimulation of human neurons leads to a decrease in the percentage infected with *T. gondii*.** Human neurons cultures were stimulated with human IFN-γ (100 U/ml) or vehicle for 24 h, followed by infection with listed *T. gondii* strains. At listed time points, cultures were fixed, stained with anti-Tuj1 antibodies and DAPI, and analyzed on an Operetta® CLS™ high content analysis microscope. **A** Representative images of human neurons infected with type II parasites. Green = anti-Tuj1 antibodies, red = mCherry expressing parasites, blue = DAPI. Scale bar = 20 μm. **B** Graph of the rate of type II infected neurons at the listed time points, normalized to the unstimulated culture. **C** As in **A** except for type III parasites. **D** As in **B** except for type III parasites. **B**, **D** Bars mean ± SD. *N* = 1100–1450 neurons/*T. gondii* strain/experiment, 2 independent differentiations. 2-way ANOVA with Sidak's multiple comparisons. Source data are provided as a Source Data file.

by neurons affects CNS parasite control[26]. This work showed that murine CNS cyst burden varied depending on neuron expression of a specific MHC I allele but did not determine if the changes in cyst burden arose from cytolytic clearance of infected neurons; induction of IFN-γ secretion by CD8+ T cells; or even non-cytolytic, perforin-dependent clearance of parasites by CD8+ T cells[57]. Our data that infected neurons clear intracellular parasites in vivo (Fig. 5E) establish that a non-cytolytic mechanism of clearance occurs and our in vitro work (Figs. 2–4) suggests that IFN-γ-dependent, cell-intrinsic clearance is more likely. Finally, as the Cre reporter mice we use only express the class I MHC H-2 L^b, these data suggest that productive CD8+ T cell-neuron interactions also occur in wild type B6 mice.

In human neurons, the data presented here show that IFN-γ-stimulation leads to a ~50% reduction in the neuron infection rate, which is consistent with the rate of clearance marker loading on PVs in IFN-γ-stimulated primary human umbilical vein endothelial cells[58].

Currently, we cannot distinguish between an IFN-γ-dependent invasion defect or clearance of intracellular parasites or both. Though we cannot discriminate between these possibilities, the IFN-γ-dependent, anti-parasite effect appears to have a more robust effect on hPSC neuron infection rates compared to murine neurons (50 vs. 25%). What mechanisms underlie this impressive IFN-γ-dependent resistance to *T. gondii* infection will be the subject of future studies.

Though we have shown that IFN-γ-stimulated neurons can clear parasites in an IRG-dependent manner, our data also suggests that major cell-specific differences in IRG efficiency exist. We found that in vitro, ~ 20–25% of intracellular parasites will be cleared by neurons in an IFN-γ, IRG-dependent manner, while other groups have shown that IFN-γ stimulated murine astrocytes, macrophages, and fibroblasts have higher rates of Irga6+ loading (50–75%) and clearance over a much shorter time (1–2 h post-infection)[33,59,60]. While some of the difference may be secondary to technical differences (e.g., use of antibody vs.

transfection of Irga6-tagged with GFP, different MOIs), part of the difference is likely secondary to a blunted cell-intrinsic immune response from neurons (as suggested by the undetectable levels of baseline STAT1). Other, not mutually exclusive possibilities include that IRG-clearance differs between neuronal subcellular locations (i.e., it might be expensive to put IRG-machinery along the whole neuron) and/or that full neuron responses require direct interactions with other cell types such as astrocytes or T cells.

Our in vivo data using II-GCre parasites are consistent with the possibility that other cell types influence neuronal clearance of intracellular parasites. In vivo we found that ~50% of II-GCre-triggered GFP+ neurons do not harbor parasites (Fig. 5). This rate of neuronal clearance is approximately double what we observed in vitro. In the in vivo setting, neurons are in constant communication with other cell types (e.g. astrocytes, microglia, infiltrating T cells), which may potentiate IRG-dependent clearance or initiate complementary methods for clearing intracellular parasites (e.g. CD40-dependent xenophagy[61,62]). Understanding why neurons incompletely clear parasites and how to improve this clearance is essential for developing curative anti-*T. gondii* therapies. Future studies will focus on mechanistically defining these in vivo vs. in vitro differences and what inhibits neuron clearance.

In summary, our findings offer substantial evidence that IFN-γ pre-stimulation enables murine and human neurons to mount anti-parasitic defenses against *T. gondii*. While much work is left to be done to understand these anti-parasitic defenses, the work presented here suggests that *T. gondii's* persistence in neurons is not simply a foregone conclusion.

## Methods

### Parasite maintenance
The parasite strains used in this study were maintained through serial passage in human foreskin fibroblasts (HFFs) using DMEM, supplemented with 10% fetal bovine serum (FBS), 2 mM Glutagro and 100 IU/ml penicillin and 100 μg/ml streptomycin. Except for the type II (Prugniaud) strain that expresses Gra16::Cre (II-GCre), the *T. gondii* strains used have been previously described[15,27].

### Generation of GCre parasites
To generate Gra16::Cre *T. gondii* parasites, we used standard molecular methods[27]. In brief, we modified a previously described plasmid—that expresses the selectable hypoxanthine xanthine guanine phosphoribosyl transferase (HPT) marker and an epitope tagged rhoptry protein fused to an NLS and Cre recombinase (*ptoxofilin-Cre*)[28]—by inserting the 5' UTR (1000 bp) and coding DNA sequences (CDS) for dense granule protein 16 (pGra16-Cre) in front of the Cre CDS using standard In-Fusion cloning (Takara Bio, 638918). The primers are listed in Supplementary Table 1. The parental type II strain type (PruΔhpt) was electroporated with the *pGra16-Cre* plasmid after linearization upstream of the relevant expression cassettes. The parasites were then subjected to several rounds of selection for HPT expression using medium containing 25 μg/ml mycophenolic acid and 50 μg/ml xanthine followed by cloning by limiting dilution[63]. Single cell clones that were HPT+ and confirmed to express the appropriate Gra16 fusion protein were then tested for efficacy in causing Cre-mediated recombination in a Cre-reporter cell line.

### Mice
All procedures and experiments were carried out in accordance with the Public Health Service Policy on Human Care and Use of Laboratory Animals and approved by the University of Arizona's Institutional Animal Care and Use Committee (#12-391). All mice were bred and housed in specific-pathogen-free University of Arizona Animal Care facilities with a 14 hr/10 hr light/dark cycle, with ambient temperature between 68° and 75° F, and 30–70% humidity. Cre reporter mice[64] (#007906) were originally purchased from Jackson Laboratories. Breeding pairs of *Irgm1/m3*−/− (Irgm1/3 KO) mice[50] were generously provided by Greg Taylor (Duke University, Durham, NC).

### In vivo infection with GCre parasites
Cre reporter mice were intraperitoneally infected with 10,000 or 20,000 freshly lysed Gra16-Cre (II-GCre) parasites. Brains were harvested at 21 dpi followed by drop fixation in 4% PFA at 4 °C overnight before being transferred and stored in 30% sucrose until they were sectioned. *T. gondii* infected, sucrose embedded brains were sagittally sectioned to 200 μm on a vibratome and stored in cryoprotectant media (0.05 M sodium phosphate buffer containing 30% glycerol and 30% ethylene glycol). Sections were cleared using a modified PACT clearing protocol previously described[53]. In brief, sections were incubated overnight in a hydrogel monomer solution at 4 °C and deoxygenated the next morning by bubbling nitrogen gas into sample vials. Samples were incubated at 42 °C to initiate crosslinking of proteins, washed, and incubated in 8% sodium dodecyl sulfate (SDS) at 45 °C to remove lipids. After multiple wash steps to remove SDS from the sections, nuclei were stained using Hoechst. Samples were then washed and submerged overnight in sorbitol refractive index matching solution (sRIMS) consisting of 70% Sorbitol and 0.01% $NaN_3$ dissolved in 0.02 M PB, followed by mounting on spacer slides in fresh sRIMS. Mounted sections were then imaged at 40× magnification using a Zeiss NLO 880 confocal microscope (Imaging Core—Marley, University of Arizona)[54]. We created z-stack tile scans of each neuron to ensure capturing as much of the neuron's axon and processes as were available in each section. Stitched images were converted to Imaris files and imported into Bitplane Imaris software (IMARIS Software X64; Version 9.7.2), where neuronal projections were rendered using the filaments tool.

### Primary murine neuron cultures
Primary neurons were cultured by previously described methods with minor modifications[65]. Culture plates were coated overnight with either 0.001% poly-L-lysine (Sigma, Cat # P4707) solution for plastic surfaces or 100 μg/ml poly-L-lysine hydrobromide (Sigma, Cat # P6282) for glass surfaces. Neurons were seeded in plating media, which is MEM (Thermo Fisher, Cat #11095080) supplemented with 0.6% D-glucose (Sigma, Cat # G8270), L-Glutamine (Thermo Fisher, Cat # 25030024), and 5% fetal bovine serum (FBS) (Thermo Fisher, Cat # 26140079) at appropriate densities: 500,000 in 6-well plates for RNA and protein extraction, 100,000 in 24 well plates with coverslips for imaging, and 20,000 in 96 well plates for counting. Four hours after plating, the media was changed to complete neurobasal media, which consists of Neurobasal™ media (Thermo Fisher, Cat # 21103049) supplemented with B27 (Thermo Fisher, Cat # 17504001), L-Glutamine and Penicillin/Streptomycin (Thermo Fisher, Cat # 15070063) antibiotic cocktail. On day in vitro (DIV) 4, neurons received a half volume media change of complete neurobasal media with 5 μM cytosine arabinoside to stop glial proliferation. Thereafter, one third media exchanges with complete neurobasal media occurred every 3-4 days. All the experiments were performed on 12 DIV neurons.

### IFN-γ stimulation and *T. gondii* infection
Primary neurons were pre-stimulated with 100 U/ml of mouse recombinant IFN-γ (R&D Systems, Cat # 485-MI-100) for 4 & 24 h (RNA extraction), 24 & 48 h (protein extraction), or 24 h (*T. gondii* infections). Freshly syringe-lysed *T. gondii* parasites resuspended in complete neurobasal media were used to infect the primary neurons at MOI = 4 (for 3 hpi), MOI = 2 (For 12 hpi), MOI = 1 (for 24 hpi) and MOI = 0.2 (for 48 and 72 hpi) time points.

## Quantitative real time PCR

For gene expression quantification, RNA was extracted from 4 h and 24 h IFN-γ stimulated primary neurons using TRIzol™ Reagent (Life Technologies) and following the manufacturer's protocol. 500 ng of total RNA was converted into first strand cDNA using a High-Capacity cDNA Reverse Transcription Kit (Applied Biosystems™; Cat No: 4368814) and following the manufacturer's instructions. Using the primers listed in Supplementary Table S1, IFN-γ response and IRG pathway genes were amplified using SYBR green fluorescence detection with an Epppendorf Mastercycler ep realplex 2.2 system. GAPDH was used as a housekeeping gene to normalize RNA levels. Results were calculated using the $2^{-\Delta\Delta CT}$ method[66].

## Protein extraction and Western blotting

Primary neurons were either unstimulated or stimulated with IFN-γ for 24 h and 48 h, followed by total protein extraction as previously described[67]. Equal amounts of protein were subjected to SDS-PAGE, transferred to PVDF membrane and western blotting was done by standard methods. The blots were imaged using the Odyssey Infrared Imaging Systems (LI-COR Biosciences) and quantification was done using Image Studio software (Version 5.2).

## Murine neuronal clearance assay

Primary neurons (wild type or Irgm1/3 KO) plated were plated on a poly-L-lysine coated 96-well plate (20,000/well) and were either unstimulated or pre-stimulated with IFN-γ for 24 h prior to infection with mCherry-expressing *T. gondii* parasites. The cells were then fixed and labeled with anti-NeuN antibodies (neuronal nuclei) and DAPI (all nuclei). Wells were imaged using an Operetta® CLS™ high content analysis system (Functional Genomics Core, University of Arizona). Generated images were then analyzed using Image J software (NIH, Version 1.52 K). All results were normalized to the neuron infection rate in the unstimulated group.

For assays involving II-GCre parasites, neurons from either wild type (Cre reporter) or Cre reporter Irgm1/3 KO mice were plated on poly-L-lysine coated 6 well plates (500,000/well). At the appropriate DIV, the neurons were either stimulated with IFN-γ or vehicle for 24 h followed by infection with II-GCre parasites. At 72 hpi, the plates were processed as described above. Fixed and stained plates were then analyzed using an epifluorescent microscope (EVOS). The person analyzing the images was blinded to IFN-γ stimulation and/or infecting parasite strain.

## Immunofluorescence assay

Cells were grown on poly-L-lysine-coated glass coverslips (described above) and were processed by methods as previously described[65].

## Antibodies

The following primary antibodies were used in the study: mouse anti-tubulin beta III isoform (Tuj1),clone TU20 (MAB1637, Millipore, 1:1000); rabbit anti-β3-Tubulin, D71G9 (similar to Tuj1) (5568 S, CST, 1:1000); mouse anti-NeuN clone A60 (MAB377, Millipore, 1:1000); rabbit anti-Glial Fibrillary Acidic protein (GFAP) (Z0334, DAKO, 1:500); rabbit anti-S100 (Z0311, DAKO, 1:500); rabbit anti-ALDH1L1 (Ab87117, Abcam, 1:500); chicken anti-Iba1 (Ab 139590, Abcam, 1:500), rabbit anti-STAT1 (Ab47425, Abcam; 1:500); mouse anti-pSTAT1 pY701 clone14/p-STAT1 (612132, BD Biosciences, 1:250); rabbit anti-pSTAT1 Tyr701, Clone 58D6 (9167, CST, 1:200); rabbit anti-*T. gondii* polyclonal antibody (PA-17252, ThermoFisher, 1:5000); mouse anti-SAG1 DG52 (gift John Boothroyd, 1:10,000)[68]; mouse anti-SRS9 (gift John Boothroyd, 1:10,000) (Kim and Boothroyd, 2005); mouse anti-Irga6 (1:1500), mouse anti-Irgb6 (1:250) (gift Jonathan Howard)[33]; rabbit ant-HA C29F4 (3724 S, CST, 1:500); mouse anti-ROP2/4 (1:1000, gift John Boothroyd)[69]; mouse anti-SAG2X (1:2000, gift John Boothroyd)[70]; DAPI (D3571, Thermo Fisher, 1:1000); Hoechst 33342 Trihydrochloride,

Trihydrate (H3570, Thermo Fisher, 1:1000). The following species-appropriate secondary antibodies were used: Alexa Fluor 405 goat anti-rabbit IgG, Alexa Fluor 488 goat anti-mouse IgG, Alexa Fluor 568 goat anti-rabbit IgG, Alexa Fluor 647 goat anti-mouse IgG, Alexa Fluor 647 goat anti-chicken IgG, Alexa Fluor 647 goat anti-rabbit IgG, donkey anti-mouse IgG, DyLight 680 conjugate (1:10000), and donkey anti-rabbit IgG, DyLight 800 conjugate (1:10000). Unless otherwise noted, secondary antibodies were obtained from Life Technologies and used at a concentration of 1:500.

## Human pluripotent stem cell derived neurons

Human embryonic neural stem cells (NSCs) derived from the NIH-approved H7 embryonic stem cells (WiCell WA07) were purchased from the University of Arizona iPSC core (https://stemcells.arizona.edu/). All procedures and experiments were approved by the University of Arizona's Embryonic Stem Cell Research Oversight (ESCRO) Committee. The NSCs were expanded and differentiated into cortical layer neurons using a previously described protocol[71] with minor modifications. Briefly, NSCs were expanded on Matrigel® Matrix (Corning®, #354277) coated plates using NSC expansion medium (NEM) (Thermofisher, Cat # A1647801). The media was changed every other day until NSCs reached confluence. For differentiation to cortical neurons, P2 passaged NSCs were plated on poly-L-ornithine (20 μg/ml) (Sigma, Cat # P4957) and laminin (5 ug/ml) (Thermo Fisher, Cat # 23017015) coated plates and differentiated for 14 days, using neural differentiation medium (NDM), which consists of Neurobasal™ medium, 2 mM L-Glutamine, 1% B-27, 200 μM L-Ascorbic acid (Sigma, Cat # A92902), 0.5 mM c-AMP (Stem Cell Technologies Cat # 73886), 20 ng/ml BDNF (Stem Cell Technologies Cat # 78005), 20 ng/ml GDNF (Stem Cell Technologies Cat # 78058), 20 ng/ml NT-3 (Stem Cell Technologies Cat # 78074), and Penicillin/Streptomycin cocktail. The culture medium was exchanged with fresh NDM every 2–3 days.

## Human neuronal clearance assay

The clearance assay in human neurons was performed as described for murine neurons except that human IFN-γ (R&D Systems, Cat # 285-MI-100) was used for pre-stimulation.

## Statistical analyses

Graphs were generated and statistical analyses were performed using Graphpad Prism 9.1.2 software. The specific test used (e.g., ANOVA vs. *t*-test) is noted in each Figure.

## Reporting summary

Further information on research design is available in the Nature Research Reporting Summary linked to this article.

## Data availability

Source data for Figs. 1–6 and Supplementary Figs. 1-3 are provided with this paper. Source data are provided with this paper.

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

## Acknowledgements

We would like to thank all members of the Koshy Lab for helpful dis-
cussions, and especially Kaitlin Thomas for her critical review of the
manuscript. We would like thank Greg Taylor (Duke University), for
providing us with breeding pairs for Irgm1/3 KO mice and Jonathan
Howard (Instituto Gulbenkian de Ciência), for providing us with anti-
Irga6 antibodies. We offer many thanks to Dr. Jared Churko (University of
Arizona) and Dr. Rita Sattler (Barrow Neurological Institute) and their lab
members for their support with human neuron differentiations. We also
thank Dr. Chris Hunter for his careful reading and editing of this manu-
script. Funding was provided by the National Institutes of Health
(NS095994 (A.A.K.); AI157247 (AAK co-PI); AI147711 (J.A.K.)) and the BIO5
Institute, University of Arizona (A.A.K., S.C.). The funders had no role in
study design, data collection and analysis, decision to publish, or pre-
paration of the manuscript.

## Author contributions

S.C., J.A.K., E.F.M. designed and performed experiments, validated
methodologies, curated data, and analyzed data. S.C. wrote the original
draft of the paper with J.A.K. and E.F.M. contributing specific, relevant
sections. AAK supervised the project and edited the paper.

## Competing interests

The authors have declared that no competing interests exist.

## Additional information

**Supplementary information** The online version contains supplementary
material available at

Anita A. Koshy.

**Peer review information** *Nature Communications* thanks Ildiko Dunay
and the other, anonymous, reviewer(s) for their contribution to the peer
review of this work. Peer reviewer reports are available.

