## [Peer Review File · Nature Communications]

Reviewers' comments:

Reviewer #1 (Remarks to the Author):

It was previously shown that "immunologically incompetent" neurons are the strategic location in the CNS where the intracellular parasite *Toxoplasma gondii* persist. Early studies applied IFN- γ stimulation to prove that neurons are not able to clear the parasites directly.

In the current study Chandrasekaran et al. stimulated primary murine neurons with IFN- γ to characterize neuronal immune responses. They isolated neurons from wild-type and transgenic mice and infected them in vitro with parental and transgenic parasites to reconsider IFN- γ dependent neuronal parasite clearance. With the similar mechanism as macrophages, IFN- γ stimulated neurons cleared the parasites via immunity regulated GTPases. Next, neurons were isolated from infected mice, resulting in neuronal GFP expression after full invasion, suggested that about 40% of the *T. gondii*-invaded neurons eliminated the parasites. Also, human stem cell derived neurons reduced parasite infection rate by about 50% upon IFN- γ stimulation. This study describes the first time that neurons are able to eliminate *T. gondii* in an IFN- γ dependent manner. The results are very interesting and significant for the *Toxoplasma* field. The study is well designed, methodology is sound, the data analysis and interpretation is correct.

Minor:

Fig.1C the x axis is difficult to read

Fig.1D was repeated only 2 times observing 48-51 cells, thus one more independent experiment is suggested

Fig.3 needs to be redesigned as it is difficult to read.

Figures legends contain some typos

Reviewer #2 (Remarks to the Author):

In the present study, Koshy and colleagues have analyzed immune responses in neurons against an important human and animal intracellular pathogen, *Toxoplasma*. They developed a new transgenic parasite line expressing Cre fused with a dense granule protein, allowing them to identify *Toxoplasma*-infected/post-infected in the Cre reporter cells or mice. Using the Cre-expressing *Toxoplasma*, they found that neurons in mice expressed GFP without the parasites, suggesting that parasites have infected

but been cleared. They further analyzed the mechanisms of parasite clearance and found that interferon- γ -inducible GTPases such as IRGs and GBPs are important for IFN- γ -mediated parasite clearance from neurons. Furthermore, the authors also found that IFN- γ stimulated human neurons clear the parasite.

(Major point)

The authors described “*Toxoplasma gondii* persists in neurons because neurons cannot clear intracellular parasites,,,” as dogma. However, given that IFN- γ -dependent inhibition of the parasite proliferation in human neurons has been already reported (PMID: 31119110), the IFN- γ -induced anti-*Toxoplasma* immune response in neurons described in the present study is conceptually no longer novel.

(Minor point)

- 1) The authors should describe the strategy to generate Gra16-Cre *Toxoplasma* in Methods. The Cre possessed NLS signal? How the Cre fused with Gra16? What is the drug resistance marker for selection?
- 2) Fig. 5; the authors discriminate infected or uninfected neurons only by GFP-negative “hole” formation with Hoechst. The authors should have tested by indirect immuno-staining parasite proteins for the presence in the “hole”.
- 3) Fig. 3; what is IRG KO? *Irgm1/m3* KO?
- 4) Fig. S3; what is *Pru::TCre*? Is this R-Cre?
- 5) Fig. S4; the generation of mice is not worthwhile making a figure.

Reviewer #3 (Remarks to the Author):

Major findings

The study demonstrates that primary murine neurons can effectively control the intracellular parasite *T. gondii* after stimulation with IFN γ . This finding clarifies previous conflicting results in the literature about the role of IFN γ in controlling *T. gondii* infection within neurons. It also extends previous studies demonstrating that neurons stimulated with interferons can control viral infections to now include protozoan parasites.

The study is largely confirmatory of mechanisms that have been demonstrated in other murine cells. The authors provide some evidence that clearance in IFN γ stimulated neurons is due to expression and recruitment of IRGs. This is not surprising as the IRG system constitutes the major mechanism of control in T gondii in IFN γ stimulated cells of multiple lineages in rodents.

Major points

1) The reduction in neuronal infection after IFN γ stimulation is modest – showing only 25% reduction at 24 hrs. This result contrasts with other cell types including astrocytes where clearance rates are much more dramatic. So although the findings suggest neurons can exert some control over infection after IFN γ stimulation, it is not clear how important the role of neuron control of infection is in vivo – despite some evidence that it can occur (Fig 5).

2) The authors also show modest control of intracellular T gondii in human neurons treated with IFN γ , although there is no insight into the mechanism of clearance. This greatly diminishes the significance of the findings that remain largely observational.

3) The dense granule protein being used as a fusion should be referred to by name in the text – rather than being anonymous. The export of this protein is somewhat surprising as fusion of globular proteins to GRAs normally completely blocks export (and ablates function). The authors should provide some estimate of how effectively this protein gets exported- independent of the Cre mediated activation of GFP. The large percentage of GFP+ cells that are not infected suggest the background is rather high in this assay. Although the authors suggest that this is due to division of fibroblasts after infection, the evidence for this is indirect. If the authors are convinced that the II-GCre line only marks infected neurons, they should present these data and omit the confusing results that occur in dividing cells.

4) The use of type III parasites expressing ROP18 and Irgm1/3 KO neurons provides partial evidence for the role of the IRG system in the control of T gondii infection in neurons. However, these experiments are lacking an important control. Do IFN γ treated neurons clear type III parasites that do not express ROP18? Such data should be included in Fig 4.

Point-by-Point Response for Reviewer Remarks to NCOMMS-21-46844

Reviewer #1

It was previously shown that "immunologically incompetent" neurons are the strategic location in the CNS where the intracellular parasite *Toxoplasma gondii* persist. Early studies applied IFN- γ stimulation to prove that neurons are not able to clear the parasites directly.

In the current study Chandrasekaran et al. stimulated primary murine neurons with IFN- γ to characterize neuronal immune responses. They isolated neurons from wild-type and transgenic mice and infected them in vitro with parental and transgenic parasites to reconsider IFN- γ dependent neuronal parasite clearance. With the similar mechanism as macrophages, IFN- γ stimulated neurons cleared the parasites via immunity regulated GTPases. Next, neurons were isolated from infected mice, resulting in neuronal GFP expression after full invasion, suggested that about 40% of the *T. gondii*-invaded neurons eliminated the parasites. Also, human stem cell derived neurons reduced parasite infection rate by about 50% upon IFN- γ stimulation. This study describes the first time that neurons are able to eliminate *T. gondii* in an IFN- γ dependent manner. **The results are very interesting and significant for the *Toxoplasma* field.** The study is well designed, methodology is sound, the data analysis and interpretation is correct.

*We appreciate Reviewer #1's recognition that we are significantly changing the current *Toxoplasma*-neuron paradigm.*

Minor:

Fig.1C the x axis is difficult to read

Thank you for pointing this out, we have made the font bigger and bolded it.

Fig.1D was repeated only 2 times observing 48-51 cells, thus one more independent experiment is suggested-

We have repeated this a third time and added the data.

Fig.3 needs to be redesigned as it is difficult to read-

We have re-designed the figure and we hope it is now easier to read/understand.

Figures legends contain some typos

We apologize for not catching the typos and believe we have now fixed all of them.

Reviewer 2

In the present study, Koshy and colleagues have analyzed immune responses in neurons against an important human and animal intracellular pathogen, *Toxoplasma*. They developed a new transgenic

parasite line expressing Cre fused with a dense granule protein, allowing them to identify Toxoplasma-infected/post-infected in the Cre reporter cells or mice. Using the Cre-expressing Toxoplasma, they found that neurons in mice expressed GFP without the parasites, suggesting that parasites have infected but been cleared. They further analyzed the mechanisms of parasite clearance and found that interferon- γ -inducible GTPases such as IRGs and GBPs are important for IFN- γ -mediated parasite clearance from neurons. Furthermore, the authors also found that IFN- γ stimulated human neurons clear the parasite.

We appreciate this summary.

1. The authors described “Toxoplasma gondii persists in neurons because neurons cannot clear intracellular parasites,,,” as dogma. However, given that IFN- γ -dependent inhibition of the parasite proliferation in human neurons has been already reported (PMID: 31119110), the IFN- γ -induced anti-Toxoplasma immune response in neurons described in the present study is conceptually no longer novel.

Reviewer 2’s major concern is that the novelty of this submission is compromised because “IFN- γ -induced anti-Toxoplasma immune response in [human] neurons” has already been described (PMID: 31119110). In this context, it is important to recognize two things about Bando et al Front Cell Infect Micro 2019:

1. The paper uses a variety of cell lines to model the role of IFN- γ +/- IL-1B in controlling T. gondii. One of these cell lines is a neuroblastoma cell line, which the paper cites as a CNS cell line. Neuroblastomas are embryonal tumor cells of the autonomic nervous system and the majority arise in the adrenal glands that sit above the kidney. It’s a simple mistake- but these are not equivalent to cortical neurons.

*2. The authors did use primary human neurons but these cells were co-cultured with **infected** monocytes- i.e., the neurons were not directly infected. In fact, it is unclear if the neurons ever became infected. In short, the authors do not assess the impact of their stimuli on the ability of neurons to control growth of T. gondii. Thus, to the best of our knowledge, our work is the only study that **directly and unequivocally** addresses the effect of IFN- γ on human neuron defenses against T. gondii. We did not cite this paper previously, but we have now cited the paper (lines 318-324) and outlined the differences between the prior work and the current submission.*

Two final points in terms of novelty. 1. To the best of our knowledge, since the study in 2001 (Schluter et al), there have been no studies that directly address the role of IFN- γ -dependent control of T. gondii in infected neurons. As evidence that the lack of follow up shows that the wider T. gondii community accepts that cytokine-stimulated neurons do not control intracellular parasites we offer the recent review by two prominent senior T. gondii researchers (Ildiko Dunay and David Sibley), which states “Neurons activated with IFN- γ are not able to clear the parasites in vitro (2001 study)- perhaps due to decreased or different transcriptional responses to IFN- γ ” (Matta et al Nat Micro Review 2021). A review by Luder and Rahman states the same idea (pg 205, 2nd column, Microbial Cell 2017). Thus, our results showing IFN- γ , cell-intrinsic control of T. gondii challenge this long held assumption.

To be clear, we are not saying that the data provided Schluter et al 2001 was wrong. In fact, we have now included data using the RH strain and find the same result as Schluter et al 2001 (see updated Fig 2A). Here we provide experimental evidence that explains how those authors reached their conclusion based on the parasite strain used and highlight that the more common, less virulent strains are susceptible to cell-intrinsic, IFN- γ -dependent neuronal defense mechanisms. Thus, our work resolves these differences.

2. A recent paper by the Blanchard group indicates that MHC class I on neurons helps with parasite control (Salvioni et al Cell Reports 2019) but it is unclear whether that control is secondary to cytolysis of infected neurons; induction of IFN- γ ; or even non-cytolytic perforin-dependent clearance by CD8 T

cells (Suzuki et al 2010). Our data showing clearance infected neurons in vivo (Fig 5) establish that a mechanism of control that is independent of cytotoxicity of infected neurons exists and that IFN- γ -dependent, cell-intrinsic clearance is more likely. In addition, unlike the Salvioni paper, the Cre reporter mice we use only express the class I MHC H-2 L^d. Our data, therefore, are the first to suggest that productive CD8⁺ T cell-neuron interactions also occur in wild type B6 mice. We have added these important points to the discussion.

(Minor point)

1) The authors should describe the strategy to generate Gra16-Cre Toxoplasma in Methods. The Cre possessed NLS signal? How the Cre fused with Gra16? What is the drug resistance marker for selection? We apologize for the limited way in which we described the generation of the Gra16-Cre strain. We have now provided more details about generating these parasites, including referencing the original ptoxofilin-Cre plasmid that was modified to create a pgra16-Cre plasmid and identifying the selection marker (hpt).

2) Fig. 5; the authors discriminate infected or uninfected neurons only by GFP-negative “hole” formation with Hoechst. The authors should have tested by indirect immuno-staining parasite proteins for the presence in the “hole”.

We appreciate the Reviewer’s concern. Staining 200 μ m thick brain sections with traditional antibodies is extremely challenging, especially when matched with the need to preserve intrinsic GFP signal. To address the Reviewer’s concern, we have stained thick sections with both Hoechst and anti-*T. gondii* antibodies to show that what we identify as parasites by Hoechst directly matches with anti-*T. gondii* staining (updated Fig 5 and Fig S4A, B). In addition, we have provided more examples of the correlation between Hoechst and “the hole” (Fig S4C).

3) Fig. 3; what is IRG KO? Irgm1/m3 KO? 4) Fig. S3; what is Pru::TCre? Is this R-Cre?

Thank you for catching these inconsistencies. Through various drafts we had used IRG KO and Pru::TCre to denote Irgm1/m3 KO and RCre, respectively. We thought we had caught all of these inconsistencies but clearly we missed several. We have fixed these inconsistencies.

5) Fig. S4; the generation of mice is not worthwhile making a figure.

We have removed the figure.

Reviewer 3

The study demonstrates that primary murine neurons can effectively control the intracellular parasite *T. gondii* after stimulation with IFN γ . This finding clarifies previous conflicting results in the literature about the role of IFN γ in controlling *T. gondii* infection within neurons. It also extends previous studies demonstrating that neurons stimulated with interferons can control viral infections to now include protozoan parasites.

We appreciate that Reviewer 3 acknowledges that the body of work in this paper has rigorously addressed how IFN- γ affects control of *T. gondii* within neurons, thereby resolving a long-standing issue in the field, which also has broader relevance to other neurotropic infections.

The study is largely confirmatory of mechanisms that have been demonstrated in other murine cells. The authors provide some evidence that clearance in IFN γ stimulated neurons is due to expression and recruitment of IRGs. This is not surprising as the IRG system constitutes the major mechanism of control in *T. gondii* in IFN γ stimulated cells of multiple lineages in rodents.

It is important to recognize that, like anti-viral responses, there appears to be cell-specific variability in the anti-parasitic responses in both human (Fisch et al 2019) and murine cells. For example, while in

vivo work showed that *Irgd/IRG-47* KO mice are more susceptible to *T. gondii* (Taylor et al 2001), Butcher et al 2005 showed that *IFN- γ* -dependent control of *T. gondii* in murine macrophage does not require *Irgd/IRG-47*. Given the work showing that neurons have unique responses to *IFN- γ* (Rose et al 2007) coupled with an understanding of cell-specific *IFN- γ* responses, there is no reason to expect neurons to clear intracellular parasites by the mechanism used in non-neuronal cells. The potential for neurons to use alternative mechanisms for clearance is emphasized by the fact that autophagy is particularly highly utilized in neuronal clearance of some viruses (Orvedahl et al 2010). The options for how *IFN- γ* might influence neuron control of *T. gondii* is broad [e.g., decreased attachment/invasion, killing of parasites via *NO* or tryptophan sequestration (nutritional immunity), and autophagy]. For these reasons, the data showing that *IFN- γ* leads to clearance of intracellular parasites (*GCre* parasites) by the *IRGs* are of particular importance. Basically, it's better to experimentally test and verify than to guess. This is the first time that *IRG* recruitment has been shown in neurons.

Major points

1) The reduction in neuronal infection after *IFN γ* stimulation is modest – showing only 25% reduction at 24 hrs. This result contrasts with other cell types including astrocytes where clearance rates are much more dramatic. So although the findings suggest neurons can exert some control over infection after *IFN γ* stimulation, it is not clear how important the role of neuron control of infection is *in vivo* – despite some evidence that it can occur (Fig 5).

*We agree with Reviewer 3 that the difference between in vitro IFN- γ mediated neuron clearance and other cells such as astrocytes and macrophages is notable. Where we disagree with Reviewer #3 is with the discounting of our in vivo data as lacking importance. As we note in the Discussion, the increased clearance rate of neurons in vivo (50% vs. 20-25%) suggests other factors may be at play, including other mechanisms of clearance, other cells making neuronal clearance efficient, etc. Thus, cell-specific knockouts of a single pathway in mice may not show major differences but that may be because other pathways are upregulated. Using much more expansive blockade (cell-specific *IFN- γ* receptor KOs) is likely to lead to a lack of parasite control but potentially for many reasons, not just neuron-specific clearance of intracellular pathways. In addition, we would like to highlight that if one wants to understand how to make “curative” therapies for *T. gondii*, then we need to understand why *T. gondii* persists in neurons. If we can understand why neurons are less effective than other cells at clearing *T. gondii*, then perhaps we can find ways to remove those neuron-specific blocks. Such a possibility would be akin to how understanding how *PD1* and *CTLA4* repression of T cell response had led to a revolution in cancer therapeutics. We have modified the discussion to emphasize these important implications.*

*For these reasons, like Reviewer #1, we strongly believe that having moved from a state of “*IFN- γ* stimulated neurons do not clear intracellular parasites” to clear evidence that *IFN- γ* stimulation leads to clearance of parasites in some neurons in vitro and in vivo is a substantial step to take for a single paper.*

[Redacted]

2) The authors also show modest control of intracellular *T. gondii* in human neurons treated with *IFN γ* , although there is no insight into the mechanism of clearance. This greatly diminishes the significance of the findings that remain largely observational.

*We respectfully disagree that showing that *IFN- γ* stimulated human neurons have anti-parasitic responses without identifying the mechanism diminishes the significance. Given the range of possible mechanisms, an in-depth study to define the mechanisms that may be at play (e.g., decreased attachment/invasion, killing of parasites via *NO* or tryptophan sequestration (nutritional immunity), and autophagy) is beyond the scope of this paper. To rigorously dissect out the mechanism(s) at play will be a paper unto itself.*

*In addition, many publications have shown that human non-immune cells show a rate of 5-30% for markers of clearance (Fisch et al 2019, Selleck et al 2015, Clough et al 2016). Given this body of work, a 50% reduction in neuron infection rate is quite significant for a non-immune cell, **especially as our study is the first to address the impact of IFN- γ on human neurons** (see comment to Reviewer #2).*

3) The dense granule protein being used as a fusion should be referred to by name in the text – rather than being anonymous. The export of this protein is somewhat surprising as fusion of globular proteins to GRAs normally completely blocks export (and ablates function). The authors should provide some estimate of how effectively this protein gets exported- independent of the Cre mediated activation of GFP. The large percentage of GFP⁺ cells that are not infected suggest the background is rather high in this assay. Although the authors suggest that this is due to division of fibroblasts after infection, the evidence for this is indirect. If the authors are convinced that the II-GCre line only marks infected neurons, they should present these data and omit the confusing results that occur in dividing cells.

*We apologize for this confusion. We have named GRA16 in the text. How efficiently a Gra16::fusion is exported depends on the fusion protein, which is detailed in Bracha et al bioRxiv 2018. We chose to use Gra16 as the fusion partner for Cre based upon these data (which were communicated to us personally by Lilach Sheiner). In addition, sensitive reporter systems often reveal biology not observable by more “standard” approaches. For example, until β -lactamase was fused to toxofilin (Lodoen et al 2010), it was unclear if toxofilin was released into the host cell, though this possibility was suspected (Bradley et al 2005). Similarly, the concept of *T. gondii* injecting cells without invading them was not recognized until the advent of the *T. gondii*-Cre parasites (Koshy et al 2010; Koshy et al 2012).*

The II-GCre line only marks infected neurons, which is highlighted by 98% +/- 0.62% of the GFP⁺ neurons harboring parasites when no there is no IFN- γ pre-stimulation. Conversely, for II-RCre parasites, which can inject Cre into cells without invasion, only 67% +/- 1.8% of GFP⁺ neurons harbor parasites. We removed the data with the fibroblasts (though we hated to remove primary data.)

4) The use of type III parasites expressing ROP18 and Irgm1/3 KO neurons provides partial evidence for the role of the IRG system in the control of *T gondii* infection in neurons. However, these experiments are lacking an important control. Do IFN γ treated neurons clear type III parasites that do not express ROP18? Such data should be included in Fig 4.

We have added these data.

REVIEWERS' COMMENTS

Reviewer #1 (Remarks to the Author):

This study really challenges the previous concepts and describes that neurons are able to clear *T. gondii* in an IFN- γ dependent manner. The results are very interesting and significant for the *Toxoplasma* field. The authors responded to and modified all the suggested issues thus the current updated version of the manuscript is sound.

Reviewer #2 (Remarks to the Author):

The authors sufficiently responded to my concerns raised in the previous version of the manuscript, and adequately added new data and discussion with citations. This reviewer is now persuaded that the current finding is indeed the first to show the IFN γ -mediated cell-autonomous immunity in mouse and human neurons.

Reviewer #3 (Remarks to the Author):

The authors have made a number of substantial improvements to their manuscript that address the major concerns. I am still of the opinion that: the findings that Irg proteins operate in murine neurons is not surprising, the effect on clearance by neurons is modest relative to other cell types, and the absence of mechanistic insight into how human cells clear infection limits the impact of the paper. Despite these concerns, I do agree that these data will be beneficial to the field.